# HIERARCHICAL MULTISCALE RECURRENT NEURAL NETWORKS

**Junyoung Chung, Sungjin Ahn & Yoshua Bengio** [*]
Département d'informatique et de recherche opérationnelle
Université de Montréal
{junyoung.chung,sungjin.ahn,yoshua.bengio}@umontreal.ca

## ABSTRACT

Learning both hierarchical and temporal representation has been among the long-standing challenges of recurrent neural networks. Multiscale recurrent neural networks have been considered as a promising approach to resolve this issue, yet there has been a lack of empirical evidence showing that this type of models can actually capture the temporal dependencies by discovering the latent hierarchical structure of the sequence. In this paper, we propose a novel multiscale approach, called the hierarchical multiscale recurrent neural network, that can capture the latent hierarchical structure in the sequence by encoding the temporal dependencies with different timescales using a novel update mechanism. We show some evidence that the proposed model can discover underlying hierarchical structure in the sequences without using explicit boundary information. We evaluate our proposed model on character-level language modelling and handwriting sequence generation.

## 1 INTRODUCTION

One of the key principles of learning in deep neural networks as well as in the human brain is to obtain a hierarchical representation with increasing levels of abstraction (Bengio, 2009; LeCun et al., 2015; Schmidhuber, 2015). A stack of representation layers, learned from the data in a way to optimize the target task, make deep neural networks entertain advantages such as generalization to unseen examples (Hoffman et al., 2013), sharing learned knowledge among multiple tasks, and discovering disentangling factors of variation (Kingma & Welling, 2013). The remarkable recent successes of the deep convolutional neural networks are particularly based on this ability to learn hierarchical representation for spatial data (Krizhevsky et al., 2012). For modelling temporal data, the recent resurgence of recurrent neural networks (RNN) has led to remarkable advances (Mikolov et al., 2010; Graves, 2013; Cho et al., 2014; Sutskever et al., 2014; Vinyals et al., 2015). However, unlike the spatial data, learning both hierarchical and temporal representation has been among the long-standing challenges of RNNs in spite of the fact that hierarchical multiscale structures naturally exist in many temporal data (Schmidhuber, 1991; Mozer, 1993; El Hihi & Bengio, 1995; Lin et al., 1996; Koutník et al., 2014).

A promising approach to model such hierarchical and temporal representation is the multiscale RNNs (Schmidhuber, 1992; El Hihi & Bengio, 1995; Koutník et al., 2014). Based on the observation that high-level abstraction changes slowly with temporal coherency while low-level abstraction has quickly changing features sensitive to the precise local timing (El Hihi & Bengio, 1995), the multiscale RNNs group hidden units into multiple modules of different timescales. In addition to the fact that the architecture fits naturally to the latent hierarchical structures in many temporal data, the multiscale approach provides the following advantages that resolve some inherent problems of standard RNNs: (a) computational efficiency obtained by updating the high-level layers less frequently, (b) efficiently delivering long-term dependencies with fewer updates at the high-level layers, which mitigates the vanishing gradient problem, (c) flexible resource allocation (e.g., more hidden units to the higher layers that focus on modelling long-term dependencies and less hidden units to the lower layers which are in charge of learning short-term dependencies). In addition, the learned latent hierarchical structures can provide useful information to other downstream tasks such

---

[*]Yoshua Bengio is CIFAR Senior Fellow.

as module structures in computer program learning, sub-task structures in hierarchical reinforcement learning, and story segments in video understanding.

There have been various approaches to implementing the multiscale RNNs. The most popular approach is to set the timescales as hyperparameters (El Hihi & Bengio, 1995; Koutník et al., 2014; Bahdanau et al., 2016) instead of treating them as dynamic variables that can be learned from the data (Schmidhuber, 1991; 1992; Chung et al., 2015; 2016). However, considering the fact that non-stationarity is prevalent in temporal data, and that many entities of abstraction such as words and sentences are in variable length, we claim that it is important for an RNN to dynamically adapt its timescales to the particulars of the input entities of various length. While this is trivial if the hierarchical boundary structure is provided (Sordoni et al., 2015), it has been a challenge for an RNN to discover the latent hierarchical structure in temporal data without explicit boundary information.

In this paper, we propose a novel multiscale RNN model, which can learn the hierarchical multiscale structure from temporal data without explicit boundary information. This model, called a *hierarchical multiscale recurrent neural network* (HM-RNN), does not assign fixed update rates, but adaptively determines proper update times corresponding to different abstraction levels of the layers. We find that this model tends to learn fine timescales for low-level layers and coarse timescales for high-level layers. To do this, we introduce a binary boundary detector at each layer. The boundary detector is turned on only at the time steps where a segment of the corresponding abstraction level is completely processed. Otherwise, i.e., during the within segment processing, it stays turned off. Using the hierarchical boundary states, we implement three operations, UPDATE, COPY and FLUSH, and choose one of them at each time step. The UPDATE operation is similar to the usual update rule of the long short-term memory (LSTM) (Hochreiter & Schmidhuber, 1997), except that it is executed sparsely according to the detected boundaries. The COPY operation simply copies the cell and hidden states of the previous time step. Unlike the leaky integration of the LSTM or the Gated Recurrent Unit (GRU) (Cho et al., 2014), the COPY operation retains the whole states without any loss of information. The FLUSH operation is executed when a boundary is detected, where it first ejects the summarized representation of the current segment to the upper layer and then reinitializes the states to start processing the next segment. Learning to select a proper operation at each time step and to detect the boundaries, the HM-RNN discovers the latent hierarchical structure of the sequences. We find that the straight-through estimator (Hinton, 2012; Bengio et al., 2013; Courbariaux et al., 2016) is efficient for training this model containing discrete variables.

We evaluate our model on two tasks: character-level language modelling and handwriting sequence generation. For the character-level language modelling, the HM-RNN achieves the state-of-the-art results on the Text8 dataset, and comparable results to the state-of-the-art on the Penn Treebank and Hutter Prize Wikipedia datasets. The HM-RNN also outperforms the standard RNN on the handwriting sequence generation using the IAM-OnDB dataset. In addition, we demonstrate that the hierarchical structure found by the HM-RNN is indeed very similar to the intrinsic structure observed in the data. The contributions of this paper are:

- We propose for the first time an RNN model that can learn a latent hierarchical structure of a sequence without using explicit boundary information.
- We show that it is beneficial to utilize the above structure through empirical evaluation.
- We show that the straight-through estimator is an efficient way of training a model containing discrete variables.
- We propose the *slope annealing* trick to improve the training procedure based on the straight-through estimator.

## 2 RELATED WORK

Two notable early attempts inspiring our model are Schmidhuber (1992) and El Hihi & Bengio (1995). In these works, it is advocated to stack multiple layers of RNNs in a decreasing order of update frequency for computational and learning efficiency. In Schmidhuber (1992), the author shows a model that can self-organize a hierarchical multiscale structure. Particularly in El Hihi & Bengio (1995), the advantages of incorporating a priori knowledge, "*temporal dependencies are structured hierarchically*", into the RNN architecture is studied. The authors propose an RNN architecture that updates each layer with a fixed but different rate, called a hierarchical RNN.

LSTMs (Hochreiter & Schmidhuber, 1997) employ the multiscale update concept, where the hidden units have different forget and update rates and thus can operate with different timescales. However, unlike our model, these timescales are not organized hierarchically. Although the LSTM has a self-loop for the gradients that helps to capture the long-term dependencies by mitigating the vanishing gradient problem, in practice, it is still limited to a few hundred time steps due to the leaky integration by which the contents to memorize for a long-term is gradually diluted at every time step. Also, the model remains computationally expensive because it has to perform the update at every time step for each unit. However, our model is less prone to these problems because it learns a hierarchical structure such that, by design, high-level layers learn to perform less frequent updates than low-level layers. We hypothesize that this property mitigates the vanishing gradient problem more efficiently while also being computationally more efficient.

A more recent model, the clockwork RNN (CW-RNN) (Koutník et al., 2014) extends the hierarchical RNN (El Hihi & Bengio, 1995) and the NARX RNN (Lin et al., 1996)[1]. The CW-RNN tries to solve the issue of using soft timescales in the LSTM, by explicitly assigning hard timescales. In the CW-RNN, hidden units are partitioned into several modules, and different timescales are assigned to the modules such that a module $i$ updates its hidden units at every $2^{(i-1)}$-th time step. The CW-RNN is computationally more efficient than the standard RNN including the LSTM since hidden units are updated only at the assigned clock rates. However, finding proper timescales in the CW-RNN remains as a challenge whereas our model *learns* the intrinsic timescales from the data. In the biscale RNNs (Chung et al., 2016), the authors proposed to model *layer-wise* timescales adaptively by having additional gating units, however this approach still relies on the *soft* gating mechanism like LSTMs.

Other forms of Hierarchical RNN (HRNN) architectures have been proposed in the cases where the explicit hierarchical boundary structure is provided. In Ling et al. (2015), after obtaining the word boundary via tokenization, the HRNN architecture is used for neural machine translation by modelling the characters and words using the first and second RNN layers, respectively. A similar HRNN architecture is also adopted in Sordoni et al. (2015) to model dialogue utterances. However, in many cases, hierarchical boundary information is not explicitly observed or expensive to obtain. Also, it is unclear how to deploy more layers than the number of boundary levels that is explicitly observed in the data.

While the above models focus on online prediction problems, where a prediction needs to be made by using only the past data, in some cases, predictions are made after observing the whole sequence. In this setting, the input sequence can be regarded as 1-D spatial data, convolutional neural networks with 1-D kernels are proposed in Kim (2014) and Kim et al. (2015) for language modelling and sentence classification. Also, in Chan et al. (2016) and Bahdanau et al. (2016), the authors proposed to obtain high-level representation of the sequences of reduced length by repeatedly merging or pooling the lower-level representation of the sequences.

Hierarchical RNN architectures have also been used to discover the segmentation structure in sequences (Fernández et al., 2007; Kong et al., 2015). It is however different to our model in the sense that they optimize the objective with explicit labels on the hierarchical segments while our model discovers the intrinsic structure only from the sequences without segment label information.

The COPY operation used in our model can be related to Zoneout (Krueger et al., 2016) which is a recurrent generalization of stochastic depth (Huang et al., 2016). In Zoneout, an identity transformation is randomly applied to each hidden unit at each time step according to a Bernoulli distribution. This results in occasional copy operations of the previous hidden states. While the focus of Zoneout is to propose a regularization technique similar to dropout (Srivastava et al., 2014) (where the regularization strength is controlled by a hyperparameter), our model learns (a) to dynamically determine when to copy from the context inputs and (b) to discover the hierarchical multiscale structure and representation. Although the main goal of our proposed model is not regularization, we found that our model also shows very good generalization performance.

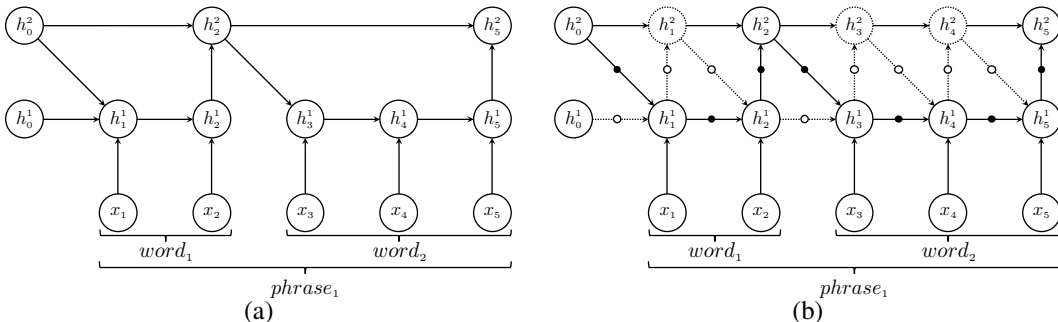

Figure 1: (a) The HRNN architecture, which requires the knowledge of the hierarchical boundaries. (b) The HM-RNN architecture that discovers the hierarchical multiscale structure in the data.

# 3  HIERARCHICAL MULTISCALE RECURRENT NEURAL NETWORKS

## 3.1  MOTIVATION

To begin with, we provide an example of how a stacked RNN can model temporal data in an ideal setting, i.e., when the hierarchy of segments is provided (Sordoni et al., 2015; Ling et al., 2015). In Figure 1 (a), we depict a hierarchical RNN (HRNN) for language modelling with two layers: the first layer receives characters as inputs and generates word-level representations (C2W-RNN), and the second layer takes the word-level representations as inputs and yields phrase-level representations (W2P-RNN).

As shown, by means of the provided end-of-word labels, the C2W-RNN obtains word-level representation after processing the last character of each word and passes the word-level representation to the W2P-RNN. Then, the W2P-RNN performs an update of the phrase-level representation. Note that the hidden states of the W2P-RNN remains unchanged while all the characters of a word are processed by the C2W-RNN. When the C2W-RNN starts to process the next word, its hidden states are reinitialized using the latest hidden states of the W2P-RNN, which contain summarized representation of all the words that have been processed by that time step, in that phrase.

From this simple example, we can see the advantages of having a hierarchical multiscale structure: (1) as the W2P-RNN is updated at a much slower update rate than the C2W-RNN, a considerable amount of computation can be saved, (2) gradients are backpropagated through a much smaller number of time steps, and (3) layer-wise capacity control becomes possible (e.g., use a smaller number of hidden units in the first layer which models short-term dependencies but whose updates are invoked much more often).

*Can an RNN discover such hierarchical multiscale structure without explicit hierarchical boundary information?* Considering the fact that the boundary information is difficult to obtain (for example, consider languages where words are not always cleanly separated by spaces or punctuation symbols, and imperfect rules are used to separately perform segmentation) or usually not provided at all, this is a legitimate problem. It gets worse when we consider higher-level concepts which we would like the RNN to discover autonomously. In Section 2, we discussed the limitations of the existing RNN models under this setting, which either have to update all units at every time step or use fixed update frequencies (El Hihi & Bengio, 1995; Koutník et al., 2014). Unfortunately, this kind of approach is not well suited to the case where different segments in the hierarchical decomposition have different lengths: for example, different words have different lengths, so a fixed hierarchy would not update its upper-level units in synchrony with the natural boundaries in the data.

## 3.2  THE PROPOSED MODEL

A key element of our model is the introduction of a parametrized boundary detector, which outputs a binary value, in each layer of a stacked RNN, and learns when a segment should end in such a way to optimize the overall target objective. Whenever the boundary detector is turned on at a time step of layer $\ell$ (i.e., when the boundary state is 1), the model considers this to be the end of a

---

[1]The acronym NARX stands for Non-linear Auto-Regressive model with eXogenous inputs.

segment corresponding to the latent abstraction level of that layer (e.g., word or phrase) and feeds the summarized representation of the detected segment into the upper layer ($\ell + 1$). Using the boundary states, at each time step, each layer selects one of the following operations: UPDATE, COPY or FLUSH. The selection is determined by (1) the boundary state of the current time step in the layer below $z_t^{\ell-1}$ and (2) the boundary state of the previous time step in the same layer $z_{t-1}^{\ell}$.

In the following, we describe an HM-RNN based on the LSTM update rule. We call this model a hierarchical multiscale LSTM (HM-LSTM). Consider an HM-LSTM model of $L$ layers ($\ell = 1, \ldots, L$) which, at each layer $\ell$, performs the following update at time step $t$:

$$\mathbf{h}_t^{\ell}, \mathbf{c}_t^{\ell}, z_t^{\ell} = f_{\text{HM-LSTM}}^{\ell}(\mathbf{c}_{t-1}^{\ell}, \mathbf{h}_{t-1}^{\ell}, \mathbf{h}_t^{\ell-1}, \mathbf{h}_{t-1}^{\ell+1}, z_{t-1}^{\ell}, z_t^{\ell-1}). \tag{1}$$

Here, $\mathbf{h}$ and $\mathbf{c}$ denote the hidden and cell states, respectively. The function $f_{\text{HM-LSTM}}^{\ell}$ is implemented as follows. First, using the two boundary states $z_{t-1}^{\ell}$ and $z_t^{\ell-1}$, the cell state is updated by:

$$\mathbf{c}_t^{\ell} = \begin{cases} \mathbf{f}_t^{\ell} \odot \mathbf{c}_{t-1}^{\ell} + \mathbf{i}_t^{\ell} \odot \mathbf{g}_t^{\ell} & \text{if } z_{t-1}^{\ell} = 0 \text{ and } z_t^{\ell-1} = 1 \text{ (UPDATE)} \\ \mathbf{c}_{t-1}^{\ell} & \text{if } z_{t-1}^{\ell} = 0 \text{ and } z_t^{\ell-1} = 0 \text{ (COPY)} \\ \mathbf{i}_t^{\ell} \odot \mathbf{g}_t^{\ell} & \text{if } z_{t-1}^{\ell} = 1 \text{ (FLUSH)}, \end{cases} \tag{2}$$

and then the hidden state is obtained by:

$$\mathbf{h}_t^{\ell} = \begin{cases} \mathbf{h}_{t-1}^{\ell} & \text{if COPY,} \\ \mathbf{o}_t^{\ell} \odot \tanh(\mathbf{c}_t^{\ell}) & \text{otherwise.} \end{cases} \tag{3}$$

Here, $(\mathbf{f}, \mathbf{i}, \mathbf{o})$ are forget, input, output gates, and $\mathbf{g}$ is a cell proposal vector. Note that unlike the LSTM, it is not necessary to compute these gates and cell proposal values at every time step. For example, in the case of the COPY operation, we do not need to compute any of these values and thus can save computations.

The COPY operation, which simply performs $(\mathbf{c}_t^{\ell}, \mathbf{h}_t^{\ell}) \leftarrow (\mathbf{c}_{t-1}^{\ell}, \mathbf{h}_{t-1}^{\ell})$, implements the observation that an upper layer should keep its state unchanged until it receives the summarized input from the lower layer. The UPDATE operation is performed to update the summary representation of the layer $\ell$ if the boundary $z_t^{\ell-1}$ is detected from the layer below but the boundary $z_{t-1}^{\ell}$ was not found at the previous time step. Hence, the UPDATE operation is executed sparsely unlike the standard RNNs where it is executed at every time step, making it computationally inefficient. If a boundary is detected, the FLUSH operation is executed. The FLUSH operation consists of two sub-operations: (a) EJECT to pass the current state to the upper layer and then (b) RESET to reinitialize the state before starting to read a new segment. This operation implicitly forces the upper layer to absorb the summary information of the lower layer segment, because otherwise it will be lost. Note that the FLUSH operation is a *hard* reset in the sense that it completely erases all the previous states of the same layer, which is different from the *soft* reset or *soft* forget operation in the GRU or LSTM.

Whenever needed (depending on the chosen operation), the gate values $(\mathbf{f}_t^{\ell}, \mathbf{i}_t^{\ell}, \mathbf{o}_t^{\ell})$, the cell proposal $\mathbf{g}_t^{\ell}$, and the pre-activation of the boundary detector $\tilde{z}_t^{\ell}$ [2] are then obtained by:

$$\begin{pmatrix} \mathbf{f}_t^{\ell} \\ \mathbf{i}_t^{\ell} \\ \mathbf{o}_t^{\ell} \\ \mathbf{g}_t^{\ell} \\ \tilde{z}_t^{\ell} \end{pmatrix} = \begin{pmatrix} \texttt{sigm} \\ \texttt{sigm} \\ \texttt{sigm} \\ \texttt{tanh} \\ \texttt{hard sigm} \end{pmatrix} f_{\texttt{slice}}\left(\mathbf{s}_t^{\text{recurrent}(\ell)} + \mathbf{s}_t^{\text{top-down}(\ell)} + \mathbf{s}_t^{\text{bottom-up}(\ell)} + \mathbf{b}^{(\ell)}\right), \tag{4}$$

where

$$\mathbf{s}_t^{\text{recurrent}(\ell)} = U_{\ell}^{\ell} \mathbf{h}_{t-1}^{\ell}, \tag{5}$$

$$\mathbf{s}_t^{\text{top-down}(\ell)} = z_{t-1}^{\ell} U_{\ell+1}^{\ell} \mathbf{h}_{t-1}^{\ell+1}, \tag{6}$$

$$\mathbf{s}_t^{\text{bottom-up}(\ell)} = z_t^{\ell-1} W_{\ell-1}^{\ell} \mathbf{h}_t^{\ell-1}. \tag{7}$$

Here, we use $W_i^j \in \mathbb{R}^{(4dim(\mathbf{h}^{\ell})+1) \times dim(\mathbf{h}^{\ell-1})}, U_i^j \in \mathbb{R}^{(4dim(\mathbf{h}^{\ell})+1) \times dim(\mathbf{h}^{\ell})}$ to denote state transition parameters from layer $i$ to layer $j$, and $\mathbf{b} \in \mathbb{R}^{4dim(\mathbf{h}^{\ell})+1}$ is a bias term. In the last layer $L$, the

---

[2] $\tilde{z}_t^{\ell}$ can also be implemented as a function of $\mathbf{h}_t^{\ell}$, e.g., $\tilde{z}_t^{\ell} = \texttt{hard sigm}(U\mathbf{h}_t^{\ell})$.

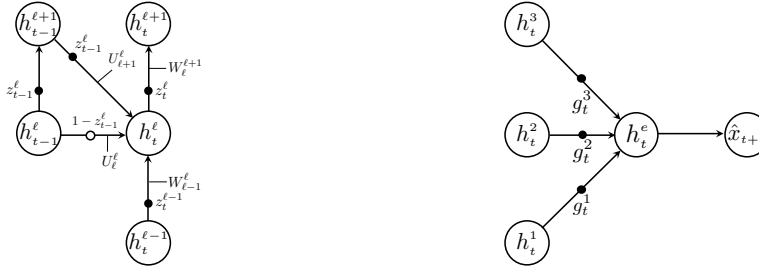

Figure 2: Left: The gating mechanism of the HM-RNN. Right: The output module when $L = 3$.

top-down connection is ignored, and we use $\mathbf{h}_t^0 = \mathbf{x}_t$. Since the input should not be omitted, we set $z_t^0 = 1$ for all $t$. Also, we do not use the boundary detector for the last layer. The `hard sigm` is defined by `hard sigm`$(x) = \max\left(0, \min\left(1, \frac{ax+1}{2}\right)\right)$ with $a$ being the slope variable.

Unlike the standard LSTM, the HM-LSTM has a top-down connection from $(\ell + 1)$ to $\ell$, which is allowed to be activated only if a boundary is detected at the previous time step of the layer $\ell$ (see Eq. 6). This makes the layer $\ell$ to be initialized with more long-term information after the boundary is detected and execute the FLUSH operation. In addition, the input from the lower layer $(\ell - 1)$ becomes effective only when a boundary is detected at the current time step in the layer $(\ell - 1)$ due to the binary gate $z_t^{\ell-1}$. Figure 2 (left) shows the gating mechanism of the HM-LSTM at time step $t$.

Finally, the binary boundary state $z_t^\ell$ is obtained by:

$$z_t^\ell = f_{\texttt{bound}}(\tilde{z}_t^\ell). \tag{8}$$

For the binarization function $f_{\texttt{bound}} : \mathbb{R} \rightarrow \{0, 1\}$, we can either use a deterministic step function:

$$z_t^\ell = \begin{cases} 1 & \text{if } \tilde{z}_t^\ell > 0.5 \\ 0 & \text{otherwise}, \end{cases} \tag{9}$$

or sample from a Bernoulli distribution $z_t^\ell \sim \text{Bernoulli}(\tilde{z}_t^\ell)$. Although this binary decision is a key to our model, it is usually difficult to use stochastic gradient descent to train such model with discrete decisions as it is not differentiable.

### 3.3 COMPUTING GRADIENT OF BOUNDARY DETECTOR

Training neural networks with discrete variables requires more efforts since the standard backpropagation is no longer applicable due to the non-differentiability. Among a few methods for training a neural network with discrete variables such as the REINFORCE (Williams, 1992; Mnih & Gregor, 2014) and the straight-through estimator (Hinton, 2012; Bengio et al., 2013), we use the straight-through estimator to train our model. The straight-through estimator is a biased estimator because the non-differentiable function used in the forward pass (i.e., the step function in our case) is replaced by a differentiable function during the backward pass (i.e., the hard sigmoid function in our case). The straight-through estimator, however, is much simpler and often works more efficiently in practice than other unbiased but high-variance estimators such as the REINFORCE. The straight-through estimator has also been used in Courbariaux et al. (2016) and Vezhnevets et al. (2016).

**The Slope Annealing Trick**. In our experiment, we use the slope annealing trick to reduce the bias of the straight-through estimator. The idea is to reduce the discrepancy between the two functions used during the forward pass and the backward pass. That is, by gradually increasing the slope $a$ of the hard sigmoid function, we make the hard sigmoid be close to the step function. Note that starting with a high slope value from the beginning can make the training difficult while it is more applicable later when the model parameters become more stable. In our experiments, starting from slope $a = 1$, we slowly increase the slope until it reaches a threshold with an appropriate scheduling.

## 4 EXPERIMENTS

We evaluate the proposed model on two tasks, character-level language modelling and handwriting sequence generation. Character-level language modelling is a representative example of discrete

| Penn Treebank | | | Hutter Prize Wikipedia | |
|---|---|---|---|---|
| **Model** | | **BPC** | **Model** | **BPC** |
| Norm-stabilized RNN | (Krueger & Memisevic, 2015) | 1.48 | Stacked LSTM (Graves, 2013) | 1.67 |
| CW-RNN | (Koutník et al., 2014) | 1.46 | MRNN (Sutskever et al., 2011) | 1.60 |
| HF-MRNN | (Mikolov et al., 2012) | 1.41 | GF-LSTM (Chung et al., 2015) | 1.58 |
| MI-RNN | (Wu et al., 2016) | 1.39 | Grid-LSTM (Kalchbrenner et al., 2015) | 1.47 |
| ME $n$-gram | (Mikolov et al., 2012) | 1.37 | MI-LSTM (Wu et al., 2016) | 1.44 |
| BatchNorm LSTM | (Cooijmans et al., 2016) | 1.32 | Recurrent Memory Array Structures (Rocki, 2016a) | 1.40 |
| Zoneout RNN | (Krueger et al., 2016) | 1.27 | SF-LSTM (Rocki, 2016b)[‡] | 1.37 |
| HyperNetworks | (Ha et al., 2016) | 1.27 | HyperNetworks (Ha et al., 2016) | 1.35 |
| LayerNorm HyperNetworks | (Ha et al., 2016) | **1.23** | LayerNorm HyperNetworks (Ha et al., 2016) | 1.34 |
| LayerNorm CW-RNN[†] | | 1.40 | Recurrent Highway Networks (Zilly et al., 2016) | 1.32 |
| LayerNorm LSTM[†] | | 1.29 | LayerNorm LSTM[†] | 1.39 |
| LayerNorm HM-LSTM | Sampling | 1.27 | HM-LSTM | 1.34 |
| LayerNorm HM-LSTM | Soft[*] | 1.27 | LayerNorm HM-LSTM | 1.32 |
| LayerNorm HM-LSTM | Step Fn. | 1.25 | PAQ8hp12 (Mahoney, 2005) | 1.32 |
| LayerNorm HM-LSTM | Step Fn. & Slope Annealing | 1.24 | decomp8 (Mahoney, 2009) | **1.28** |

Table 1: BPC on the Penn Treebank test set (left) and Hutter Prize Wikipedia test set (right). ($*$) This model is a variant of the HM-LSTM that does not discretize the boundary detector states. ($†$) These models are implemented by the authors to evaluate the performance using layer normalization (Ba et al., 2016) with the additional output module. ($‡$) This method uses test error signals for predicting the next characters, which makes it not comparable to other methods that do not.

sequence modelling, where the discrete symbols form a distinct hierarchical multiscale structure. The performance on real-valued sequences is tested on the handwriting sequence generation in which a relatively clear hierarchical multiscale structure exists compared to other data such as speech signals.

## 4.1 CHARACTER-LEVEL LANGUAGE MODELLING

A sequence modelling task aims at learning the probability distribution over sequences by minimizing the negative log-likelihood of the training sequences:

$$\min_{\theta} -\frac{1}{N} \sum_{n=1}^{N} \sum_{t=1}^{T^n} \log p\left(x_t^n \mid x_{<t}^n; \theta\right), \qquad (10)$$

where $\theta$ is the model parameter, $N$ is the number of training sequences, and $T^n$ is the length of the $n$-th sequence. A symbol at time $t$ of sequence $n$ is denoted by $x_t^n$, and $x_{<t}^n$ denotes all previous symbols at time $t$. We evaluate our model on three benchmark text corpora: (1) Penn Treebank, (2) Text8 and (3) Hutter Prize Wikipedia. We use the bits-per-character (BPC), $\mathbb{E}[-\log_2 p(x_{t+1} \mid x_{\leq t})]$, as the evaluation metric.

**Model**  We use a model consisting of an input embedding layer, an RNN module and an output module. The input embedding layer maps each input symbol into 128-dimensional continuous vector without using any non-linearity. The RNN module is the HM-LSTM, described in Section 3, with three layers. The output module is a feedforward neural network with two layers, an output embedding layer and a softmax layer. Figure 2 (right) shows a diagram of the output module. At each time step, the output embedding layer receives the hidden states of the three RNN layers as input. In order to adaptively control the importance of each layer at each time step, we also introduce three scalar gating units $g_t^\ell \in \mathbb{R}$ to each of the layer outputs:

$$g_t^\ell = \text{sigm}(\mathbf{w}^\ell[\mathbf{h}_t^1; \cdots; \mathbf{h}_t^L]), \qquad (11)$$

where $\mathbf{w}^\ell \in \mathbb{R}^{\sum_{\ell=1}^{L} dim(\mathbf{h}^\ell)}$ is the weight parameter. The output embedding $\mathbf{h}_t^e$ is computed by:

$$\mathbf{h}_t^e = \text{ReLU}\left(\sum_{\ell=1}^{L} g_t^\ell W_\ell^e \mathbf{h}_t^\ell\right), \qquad (12)$$

where $L = 3$ and $\text{ReLU}(x) = \max(0, x)$ (Nair & Hinton, 2010). Finally, the probability distribution for the next target character is computed by the softmax function, $\text{softmax}(x_j) = \frac{e^{x_j}}{\sum_{k=1}^{K} e^{x_k}}$, where each output class is a character.

| Text8 | |
|---|---|
| **Model** | **BPC** |
| *td*-LSTM (Zhang et al., 2016) | 1.63 |
| HF-MRNN (Mikolov et al., 2012) | 1.54 |
| MI-RNN (Wu et al., 2016) | 1.52 |
| Skipping-RNN (Pachitariu & Sahani, 2013) | 1.48 |
| MI-LSTM (Wu et al., 2016) | 1.44 |
| BatchNorm LSTM (Cooijmans et al., 2016) | 1.36 |
| HM-LSTM | 1.32 |
| LayerNorm HM-LSTM | **1.29** |

Table 2: BPC on the Text8 test set.

**Penn Treebank**   We process the Penn Treebank dataset (Marcus et al., 1993) by following the procedure introduced in Mikolov et al. (2012). Each update is done by using a mini-batch of 64 examples of length 100 to prevent the memory overflow problem when unfolding the RNN in time for backpropagation. The last hidden state of a sequence is used to initialize the hidden state of the next sequence to approximate the full backpropagation. We train the model using Adam (Kingma & Ba, 2014) with an initial learning rate of 0.002. We divide the learning rate by a factor of 50 when the validation negative log-likelihood stopped decreasing. The norm of the gradient is clipped with a threshold of 1 (Mikolov et al., 2010; Pascanu et al., 2012). We also apply layer normalization (Ba et al., 2016) to our models. For all of the character-level language modelling experiments, we apply the same procedure, but only change the number of hidden units, mini-batch size and the initial learning rate.

For the Penn Treebank dataset, we use 512 units in each layer of the HM-LSTM and for the output embedding layer. In Table 1 (left), we compare the test BPCs of four variants of our model to other baseline models. Note that the HM-LSTM using the step function for the hard boundary decision outperforms the others using either *sampling* or *soft* boundary decision (i.e., hard sigmoid). The test BPC is further improved with the slope annealing trick, which reduces the bias of the straight-through estimator. We increased the slope $a$ with the following schedule $a = \min\left(5, 1 + 0.04 \cdot N_{epoch}\right)$, where $N_{epoch}$ is the maximum number of epochs. The HM-LSTM achieves test BPC score of 1.24. For the remaining tasks, we fixed the hard boundary decision using the step function without slope annealing due to the difficulty of finding a good annealing schedule on large-scale datasets.

**Text8**   The Text8 dataset (Mahoney, 2009) consists of 100M characters extracted from the Wikipedia corpus. Text8 contains only alphabets and spaces, and thus we have total 27 symbols. In order to compare with other previous works, we follow the data splits used in Mikolov et al. (2012). We use 1024 units for each HM-LSTM layer and 2048 units for the output embedding layer. The mini-batch size and the initial learning rate are set to 128 and 0.001, respectively. The results are shown in Table 2. The HM-LSTM obtains the state-of-the-art test BPC 1.29.

**Hutter Prize Wikipedia**   The Hutter Prize Wikipedia (`enwik8`) dataset (Hutter, 2012) contains 205 symbols including XML markups and special characters. We follow the data splits used in Graves (2013) where the first 90M characters are used to train the model, the next 5M characters for validation, and the remainders for the test set. We use the same model size, mini-batch size and the initial learning rate as in the Text8. In Table 1 (right), we show the HM-LSTM achieving the test BPC 1.32, which is a tie with the state-of-the-art result among the neural models. Although the neural models, show remarkable performances, their compression performance is still behind the best models such as PAQ8hp12 (Mahoney, 2005) and decomp8 (Mahoney, 2009).

**Visualizing Learned Hierarchical Multiscale Structure**   In Figure 3 and 4, we visualize the boundaries detected by the boundary detectors of the HM-LSTM while reading a character sequence of total length 270 taken from the validation set of either the Penn Treebank or Hutter Prize Wikipedia dataset. Due to the page width limit, the figure contains the sequence partitioned into three segments of length 90. The white blocks indicate boundaries $z_t^{\ell} = 1$ while the black blocks indicate the non-boundaries $z_t^{\ell} = 0$.

Interestingly in both figures, we can observe that the boundary detector of the first layer, $z^1$, tends to be turned on when it sees a space or after it sees a space, which is a reasonable breakpoint to separate between words. This is somewhat surprising because the model self-organizes this structure

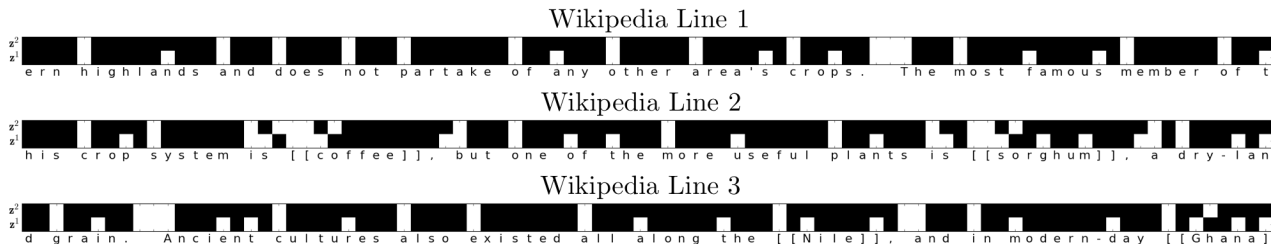

Figure 3: Hierarchical multiscale structure in the Wikipedia dataset captured by the boundary detectors of the HM-LSTM.

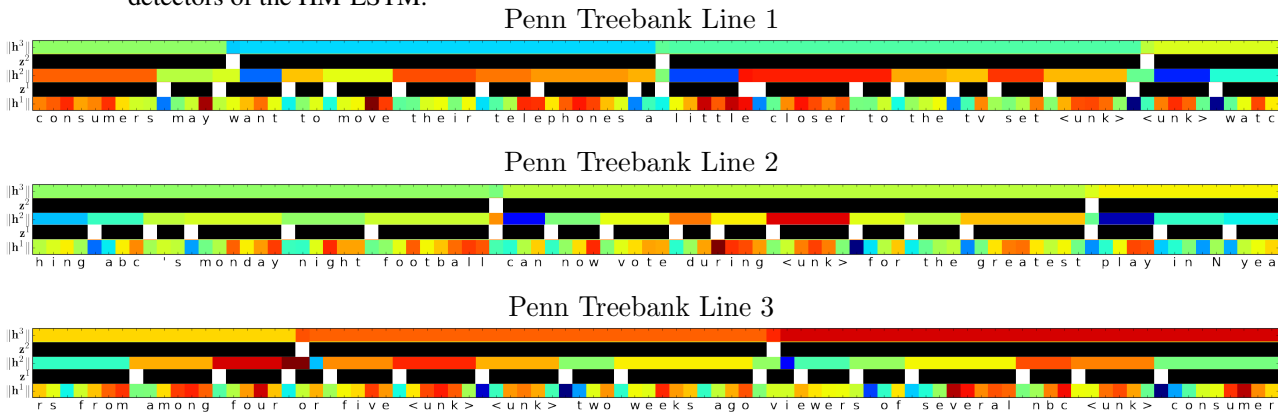

Figure 4: The $\ell^2$-norm of the hidden states shown together with the states of the boundary detectors of the HM-LSTM.

without any explicit boundary information. In Figure 3, we observe that the $z^1$ tends to detect the boundaries of the words but also fires within the words, where the $z^2$ tends to fire when it sees either an end of a word or 2, 3-grams. In Figure 4, we also see flushing in the middle of a word, e.g., "tele-FLUSH-phone". Note that "tele" is a prefix after which a various number of postfixes can follow. From these, it seems that the model uses to some extent the concept of *surprise* to learn the boundary. Although interpretation of the second layer boundaries is not as apparent as the first layer boundaries, it seems to segment at reasonable semantic / syntactic boundaries, e.g., "consumers may" - "want to move their telephones a" - "little closer to the tv set <unk>", and so on.

Another remarkable point is the fact that we do not pose any constraint on the number of boundaries that the model can fire up. The model, however, learns that it is more beneficial to delay the information ejection to some extent. This is somewhat counterintuitive because it might look more beneficial to feed the fresh update to the upper layers at every time step without any delay. We conjecture the reason that the model works in this way is due to the FLUSH operation that poses an implicit constraint on the frequency of boundary detection, because it contains both a reward (feeding fresh information to upper layers) and a penalty (erasing accumulated information). The model finds an optimal balance between the reward and the penalty.

To understand the update mechanism more intuitively, in Figure 4, we also depict the heatmap of the $\ell^2$-norm of the hidden states along with the states of the boundary detectors. As we expect, we can see that there is no change in the norm value within segments due to the COPY operation. Also, the color of $\|\mathbf{h}^1\|$ changes quickly (at every time step) because there is no COPY operation in the first layer. The color of $\|\mathbf{h}^2\|$ changes less frequently based on the states of $z_t^1$ and $z_{t-1}^2$. The color of $\|\mathbf{h}^3\|$ changes even slowly, i.e., only when $z_t^2 = 1$.

A notable advantage of the proposed architecture is that the internal process of the RNN becomes more interpretable. For example, we can substitute the states of $z_t^1$ and $z_{t-1}^2$ into Eq. 2 and infer which operation among the UPDATE, COPY and FLUSH was applied to the second layer at time step $t$. We can also inspect the update frequencies of the layers simply by counting how many UPDATE and FLUSH operations were made in each layer. For example in Figure 4, we see that the first layer updates at every time step (which is 270 UPDATE operations), the second layer updates 56 times,

| IAM-OnDB | |
| --- | --- |
| **Model** | **Average Log-Likelihood** |
| Standard LSTM | 1081 |
| HM-LSTM | 1137 |
| HM-LSTM & Slope Annealing | **1167** |

Table 3: Average log-likelihood per sequence on the IAM-OnDB test set.

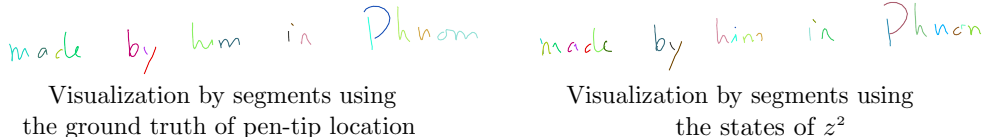

Visualization by segments using Visualization by segments using
the ground truth of pen-tip location the states of $z^2$

Figure 5: The visualization by segments based on either the given pen-tip location or states of the $z^2$.

and only 9 updates has made in the third layer. Note that, by design, the first layer performs UPDATE operation at every time step and then the number of UPDATE operations decreases as the layer level increases. In this example, the total number of updates is 335 for the HM-LSTM which is $60\%$ of reduction from the 810 updates of the standard RNN architecture.

## 4.2 HANDWRITING SEQUENCE GENERATION

We extend the evaluation of the HM-LSTM to a real-valued sequence modelling task using IAM-OnDB (Liwicki & Bunke, 2005) dataset. The IAM-OnDB dataset consists of $12,179$ handwriting examples, each of which is a sequence of $(x, y)$ coordinate and a binary indicator $p$ for pen-tip location, giving us $(x_{1:T^n}, y_{1:T^n}, p_{1:T^n})$, where $n$ is an index of a sequence. At each time step, the model receives $(x_t, y_t, p_t)$, and the goal is to predict $(x_{t+1}, y_{t+1}, p_{t+1})$. The pen-up $(p_t = 1)$ indicates an end of a stroke, and the pen-down $(p_t = 0)$ indicates that a stroke is in progress. There is usually a large shift in the $(x, y)$ coordinate to start a new stroke after the pen-up happens. We remove all sequences whose length is shorter than 300. This leaves us $10,465$ sequences for training, $581$ for validation, 582 for test. The average length of the sequences is 648. We normalize the range of the $(x, y)$ coordinates separately with the mean and standard deviation obtained from the training set. We use the mini-batch size of 32, and the initial learning rate is set to 0.0003.

We use the same model architecture as used in the character-level language model, except that the output layer is modified to predict real-valued outputs. We use the mixture density network as the output layer following Graves (2013), and use 400 units for each HM-LSTM layer and for the output embedding layer. In Table 3, we compare the log-likelihood averaged over the test sequences of the IAM-OnDB dataset. We observe that the HM-LSTM outperforms the standard LSTM. The slope annealing trick further improves the test log-likelihood of the HM-LSTM into 1167 in our setting. In this experiment, we increased the slope $a$ with the following schedule $a = \min(3, 1 + 0.004 \cdot N_{epoch})$. In Figure 5, we let the HM-LSTM to read a randomly picked validation sequence and present the visualization of handwriting examples by segments based on either the states of $z^2$ or the states of pen-tip location[3].

## 5 CONCLUSION

In this paper, we proposed the HM-RNN that can capture the latent hierarchical structure of the sequences. We introduced three types of operations to the RNN, which are the COPY, UPDATE and FLUSH operations. In order to implement these operations, we introduced a set of binary variables and a novel update rule that is dependent on the states of these binary variables. Each binary variable is learned to find segments at its level, therefore, we call this binary variable, a boundary detector. On the character-level language modelling, the HM-LSTM achieved state-of-the-art result on the Text8 dataset and comparable results to the state-of-the-art results on the Penn Treebank and Hutter Prize Wikipedia datasets. Also, the HM-LSTM outperformed the standard LSTM on the handwriting sequence generation. Our results and analysis suggest that the proposed HM-RNN can discover the latent hierarchical structure of the sequences and can learn efficient hierarchical multiscale representation that leads to better generalization performance.

---

[3]The plot function could be found at `blog.otoro.net/2015/12/12/handwriting-generation-demo-in-tensorflow/`.

ACKNOWLEDGMENTS

The authors would like to thank Alex Graves, Tom Schaul and Hado van Hasselt for their fruitful comments and discussion. We acknowledge the support of the following agencies for research funding and computing support: Ubisoft, Samsung, IBM, Facebook, Google, Microsoft, NSERC, Calcul Québec, Compute Canada, the Canada Research Chairs and CIFAR. The authors thank the developers of Theano (Team et al., 2016). JC would like to thank Arnaud Bergenon and Frédéric Bastien for their technical support. JC would also like to thank Guillaume Alain, Kyle Kastner and David Ha for providing us useful pieces of code.

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
