# Peer review of "Hierarchical Multiscale Recurrent Neural Networks"

_ICLR 2017 — accepted_

[Official Review · AnonReviewer4 · rating 8 · confidence 4 · 16 Dec 2016]
**Good paper**

This paper proposes a novel variant of recurrent networks that is able to learn the hierarchy of information in sequential data (e.g., character->word). Their approach does not require boundary information to segment the sequence in meaningful groups (like in Chung et al., 2016).

Their model is organized as a set of layers that aim at capturing the information form different “level of abstraction”. The lowest level activate the upper one and decide when to update it based on a controller (or state cell, called c). A key feature of their model is that c is a discrete variable, allowing potentially fast inference time. However, this makes their model more challenging to learn, leading to the use of the straight-through estimator by Hinton, 2012. 

The experiment section is thorough and their model obtain competitive performance on several challenging tasks. The qualitative results show also that their model can capture natural boundaries.

Overall this paper presents a strong and novel model with promising experimental results.



On a minor note, I have few remarks/complaints about the writing and the related work:

- In the introduction:
“One of the key principles of learning in deep neural networks as well as in the human brain” : please provide evidence for the “human brain” part of this claim.
“For modelling temporal data, the recent resurgence of recurrent neural networks (RNN) has led to remarkable advances” I believe you re missing Mikolov et al. 2010 in the references.
“in spite of the fact that hierarchical multiscale structures naturally exist in many temporal data”: missing reference to Lin et al., 1996

- in the related work:
“A more recent model, the clockwork RNN (CW-RNN) (Koutník et al., 2014) extends the hierarchicalRNN (El Hihi & Bengio, 1995)” : It extends the NARX model of Lin et al. 1996, not the El Hihi & Bengio, 1995.
While the above models focus on online prediction problems, where a prediction needs to be made…”: I believe there is a lot of missing references, in particular to Socher’s work or older recursive networks.
“The norm of the gradient is clipped with a threshold of 1 (Pascanu et al., 2012)”: this is not the first work using gradient clipping. I believe it was introduced in Mikolov et al., 2010.

Missing references:
“Recurrent neural network based language model.”, Mikolov et al. 2010
“Learning long-term dependencies in NARX recurrent neural networks”, Lin et al. 1996
“Sequence labelling in structured domains with hierarchical recurrent neural networks“, Fernandez et al. 2007
“Learning sequential tasks by incrementally adding  higher  orders”, Ring, 1993

[Official Review · AnonReviewer3 · rating 7 · confidence 3 · 17 Dec 2016]
**well evaluated and written paper, novel flush operation**

The paper proposes a modified RNN architecture with multiple layers, where higher layers are only passed lower layer states if a FLUSH operation is predicted, consisting of passing up the state and reseting the lower layer's state. In order to select one of three operations at each time step, the authors propose using the straight-through estimator with a slope-annealing trick during training. Empirical results and visualizations illustrate that the modified architecture performs well at boundary detection.

Pros:
- Paper is well-motivated, exceptionally well-composed
- Provides promising initial results on learning hierarchical representations through visualizations and thorough experiments on language modeling and handwriting generation
- The annealing trick with the straight-through estimator also seems potentially useful for other tasks containing discrete variables, and the trade-off in the flush operation is innovative.
Cons:
- In a couple cases the paper does not fully deliver. Empirical results on computational savings are not given, and hierarchy beyond a single level (where the data contains separators such as spaces and pen up/down) does not seem to be demonstrated.
- It's unclear whether better downstream performance is due to use of hierarchical information or due to the architecture changes acting as regularization, something which could hopefully be addressed.

[Official Review · AnonReviewer1 · rating 8 · confidence 4 · 23 Dec 2016]
**No Title**

This paper proposes a new multiscale recurrent neural network, where each layer has different time scale, and the scale is not fixed but variable and determined by a neural network. The method is elegantly formulated within a recurrent neural network framework, and shows the state-of-the-art performance on several benchmarks. The paper is well written.

Question) Can you extend it to bidirectional RNN?

[Final Decision · Program Chairs · 06 Feb 2017]
**ICLR committee final decision**

This extension to RNNs is clearly motivated, and the details of the proposed method are sensible. The paper would have benefitted from more experiments such as those in Figure 5 teasing out the representations learned by this model.